# Simultaneous Spray Drying for Combination Dry Powder Inhaler Formulations

**DOI:** 10.3390/pharmaceutics14061130

**Published:** 2022-05-26

**Authors:** Kimberly B. Shepard, Amanda M. Pluntze, David T. Vodak

**Affiliations:** Small Molecules R&D, Lonza Group AG, Bend, OR 97703, USA

**Keywords:** spray drying, dry powder inhaler, pulmonary delivery, combination therapy, lung cancer, particle engineering

## Abstract

Spray drying is a particle engineering technique used to manufacture respirable pharmaceutical powders that are suitable for delivery to the deep lung. It is amenable to processing both small molecules and biologic actives, including proteins. In this work, a simultaneous spray-drying process, termed simul-spray, is described; the process involves two different active pharmaceutical ingredient (API) solutions that are simultaneously atomized through separate nozzles into a single-spray dryer. Collected by a single cyclone, simul-spray produces a uniform mixture of two different active particles in a single-unit operation. While combination therapies for dry powder inhalers containing milled small molecule API are commercially approved, limited options exist for preparing combination treatments that contain both small molecule APIs and biotherapeutic molecules. Simul-spray drying is also ideal for actives which cannot withstand a milling-based particle engineering process, or which require a high dose that is incompatible with a carrier-based formulation. Three combination case studies are demonstrated here, in which bevacizumab is paired with erlotinib, cisplatin, or paclitaxel in a dry powder inhaler formulation. These model systems were chosen for their potential relevance to the local treatment of lung cancer. The resulting formulations preserved the biologic activity of the antibody, achieved target drug concentration, and had aerosol properties suitable for pulmonary delivery.

## 1. Introduction

Pulmonary delivery by inhalation is the preferred method of administration for the treatment of lung diseases such as asthma, COPD, pulmonary arterial hypertension, and pulmonary infections [1,2]. Inhaled treatments for additional indications such as lung cancer [3] are in clinical trials. Combination therapies delivered by dry powder inhaler (DPI) are of particular interest for improving patient experience and compliance while managing complex lung conditions [4]. To date, at least seven DPI combination therapies have been approved by the FDA (Breo Ellipta, Anoro Ellipta, Trelegy Ellipta, Utibro Breezhaler, Advair Diskus, Airduo Respiclick/Digihaler, Wixela Inhub (an Advair generic)) [5]. In all of these products, the active particles are reduced to a respirable particle size by a milling step and are then blended with carrier particles. This particle engineering approach is limited to small molecule therapies which are solid, crystalline, and insensitive to shear forces sustained during a milling process [6].

In contrast, spray drying is an enabling particle engineering technique, used to manufacture respirable particles of both small and large molecule active pharmaceutical ingredients (API) in a single-unit operation [7,8,9]. Spray drying as a pharmaceutical manufacturing technique has been reviewed extensively elsewhere [10,11]. Briefly, actives and excipients are co-dissolved in a volatile solvent and then atomized into droplets that are sprayed into a drying chamber. Heated drying gas rapidly removes the solvent, resulting in a dried powder that is collected via cyclone. The spray-drying manufacture of an inhalation formulation with more than one active compound remains challenging. Two or more APIs can be combined into a single spray solution and spray-dried such that they are molecularly mixed. Spraying together can be challenging due to the need for a common solvent, particularly when combining a biotherapeutic (which must be dissolved in an aqueous buffer to maintain biologic activity) with a low-solubility small molecule API (e.g., BCS class II or IV) that can have very low solubility in aqueous systems. In certain cases, APIs with a common solvent will not be chemically stable in the same solution or particle. Alternatively, each active could be formulated and spray-dried separately and then blended together. This approach introduces additional processing steps, potential difficulties in content uniformity, and is particularly challenging for inhalation powders, which have very poor flow and are often hygroscopic.

Another option would be to simultaneously spray two different formulation solutions into the same spray dryer through separate atomizers—a concept demonstrated by Snyder et al., who used two phosphate buffer solutions as model systems [12]. Numerous studies have demonstrated complex particle morphologies which combine two or more actives by spray-drying emulsions [13], nanoparticle suspensions [14], or other atomization technologies [15]. In contrast, simultaneous spray drying produces two distinct particle types that are intimately blended during the spray-drying process.

This study aims to demonstrate this process, hereafter termed simul-spray drying, for the first time as a method of manufacturing combination inhalation products with matching particle size distributions for DPI. More specifically, we produced spray-dried inhalation powders consisting of both a biologic and a small molecule API that also require different spray solvents. A custom atomizer wand for a lab-scale spray dryer was fabricated to accommodate two independent two-fluid atomizers, from which two solutions could be sprayed. A schematic is provided in Figure 1. The relative amounts of each spray-dried formulation present in the final product were controlled by the liquid flow rate to each atomizer and the composition of the solutions.

The model systems chosen for this study were selected due to potential relevance to the treatment of lung cancer: bevacizumab (BEV), a VEGF-inhibitor monoclonal antibody (mAb) first marketed as Avastin [16], and three small molecules commonly used in tandem with BEV treatments—erlotinib (ERL, an EGFR inhibitor), paclitaxel (PTX, a chemotherapy), and cisplatin (CP, a chemotherapy). Simul-spray drying successfully generated combination spray-dried powders with appropriate aerosol properties for lung delivery and no impact on the anti-VEGF activity of BEV. The inhalable combination formulations of these actives would not be possible by other manufacturing techniques due to bevacizumab’s incompatibility with a high-shear milling process and the lack of a common spray-drying solvent between the actives.

## 2. Materials and Methods

### 2.1. Materials

The bevacizumab drug substance was supplied as a sterile solution containing 30 mg/mL bevacizumab, 60 mg/mL trehalose, and 0.04% polysorbate 20 in 50 mM phosphate buffer at pH 6.2. Trehalose dihydrate was purchased from Pfanstiehl (Waukegan, IL, USA), and L-leucine was purchased from J.T. Baker Inc. (Phillipsburg, NJ, USA). Cisplatin was purchased from BOC Sciences (Shirley, NY, USA), paclitaxel and erlotinib were purchased from LC Laboratories (Woburn, MA, USA).

### 2.2. Spray Drying

Solutions for the spray drying of paclitaxel, cisplatin, or erlotinib were prepared by adding API and excipient solids to the solvent and stirring until dissolved. Due to their chemical instability in aqueous solutions, cisplatin solutions were used as soon as possible after preparation. The compositions are summarized in Table 1.

For the bevacizumab solution preparation, a dialysis buffer exchange was performed, as described a previous publication [17].

A customized atomizer that was fabricated for this study could introduce two solutions into the dryer simultaneously through independent nozzles. A schematic is shown in Figure 1. The atomizers are angled slightly out from one another (5–10°) such that plume interference is reduced, helping to prevent the collision and fusion of atomized droplets. The solution compositions and liquid flow rates used to manufacture the formulations in this study are listed in Table 1 and Table 2. The liquid streams were pumped into a pre-heated spray dryer with a nominal nitrogen drying gas flow rate of 500 g/min. A two-fluid nozzle was used for the atomization of each stream (Model ¼ J, with a 1650 liquid body and 64 air cap, Spraying Systems Co., Wheaton, IL, USA). The outlet temperature was set at 50 °C, and the inlet temperature was approximately 110 °C. Atomization gas pressures ranged from 15–20 psi. Spray-dried particles were collected using a 2 inch cyclone separator.

### 2.3. Drug Concentration Measurement

The drug concentration of the BEV/ERL and BEV/PTX spray-dried powders was measured using HPLC. A known mass of sample was dissolved in DMSO with sonication and analyzed on an Agilent 1100 (Agilent Technologies, Santa Clara, CA, USA), with detection at 280 nm, and then quantified against the linear standard curves of each respective active (0.05–1 mg/mL). A test solution containing known amounts of all three actives was used to confirm the specificity and accuracy of the method. A gradient method was used with a mobile phase flow rate of 1.5 mL/min and a 5 µL sample injection using an Agilent Poroshell 300 SB-C3 column at 75 °C (2.1 × 75 mm with 5 µm particles). Mobile phase A was 0.1% TFA in water, and mobile phase B was 0.1% TFA in acetonitrile, starting with 98:2 A:B for 0.1 min, then a gradient to 40:60 from 0.1 to 2 min, followed by a 0.6 min isocratic hold at 40:60 A:B before a re-equilibration back to 98:2. The total method run time including the re-equilibration step was 4.5 min, with elution of ERL, PTX, and BEV at 1.4, 1.8, and 2.1 min, respectively. ERL samples were measured in triplicate, and PTX samples were measured in quintuplicate.

BEV/CP spray-dried powders were quantitated using the 2nd derivative of the absorbance spectrum. A known mass of sample was dissolved in DMSO with sonication and analyzed using fiber optic UV-vis probes with 5 mm tips from Pion Inc. (Billerica, MA, USA) with the Au PRO software. BEV was quantitated against a linear standard curve (14–220 μg/mL) at 282 and 298 nm, identified using the software’s zero intercept mode (ZIM) as the wavelengths where CP shows no signal in the second derivative, irrespective of concentration. CP was quantitated against a linear standard curve (14–220 μg/mL) at 335–345 nm, where BEV has no absorbance. The second derivative spectra are shown in the Appendix A. A test solution containing known amounts of both actives was used to confirm the specificity and accuracy of the method. All samples were measured in triplicate.

### 2.4. Water Content

The water content of spray-dried formulations was quantified by Karl Fischer titration on a coulometric Metrohm 851 Titrando KF oven titrator (Metrohm USA Inc., Tampa, FL, USA). The generator electrode was operated in diaphragm-less mode. Sample sizes of 10–40 mg were sealed in a crimped KF vial and analyzed at an oven temperature of 105 °C and then measured in duplicate.

### 2.5. Scanning Electron Microscopy (SEM)

SEM images of spray-dried formulations were obtained using a Hitachi SU3500 SEM (Hitachi High Technologies America Inc., Schaumburg, IL, USA). Trace quantities of powder were applied to a double-sided carbon tape mounted on an aluminum stud. Samples were sputter coated with gold/palladium for approximately 6 min at 15–20 mA plasma current using a Hummer 6.2 Sputter System (Anatech+ Ltd., Battle Creek, MI, USA).

### 2.6. Thermal Analysis by Differential Scanning Calorimetry (DSC)

Thermal analysis was performed on samples using a Mettler Toledo DSC 3+ instrument (Mettler Toledo, Columbus, OH, USA). Samples were sealed in 40 µL aluminum pans, which were vented to allow moisture to boil off during the run. Samples were scanned in ADSC mode from 0 to 160 °C at 2.5 °C/min, with a modulation of 1.5 °C in amplitude every 60 s. The glass transition temperature of the samples was quantified in the Mettler Toledo STARe software from the midpoint temperatures of the transitions, measured in triplicate.

### 2.7. Aerosol Properties by Fast-Screening Impactor (FSI)

The aerosol properties of the spray-dried powders were analyzed using a fast-screening impactor device (Copley Scientific, Nottingham, UK). Based on the same technology as the pre-separator of a Next Generation Impactor, the FSI separates powders into coarse and fine particle fractions. A total of 10 mg of powder was loaded into a size 3 V Caps Plus DPI grade capsule and placed in a PlastiApe RS01 4 kPa dry powder inhaler. Using a Copley HCP5 pump and a TPK2000 unit, the test was operated at 60 L/min for 2.0 s. Particles with an aerodynamic diameter of <5 microns bypassed the impaction stage and were collected on a glass filter. The mass change of the filter before and after actuation of the DPI device was used to calculate the fine particle fraction.

### 2.8. Aerosol Particle Size Measurement

The aerodynamic particle size distribution of spray-dried powders was quantified by a TSI Aerodynamic Particle Sizer^®^ 3321 with a Model 3433 small-scale powder disperser and a Model 3302 A diluter (TSI, Shoreview, MN, USA). The air flow rate was 18.5 L/min in the powder disperser, and the sheath flow rate was 4 L/min. A 100:1 capillary was used in the diluter at a pressure of 0.32 in. of water. Distributions were measured in triplicate for each sample at 20 s each.

### 2.9. Residual Solvent Quantitation by Gas Chromatography (GC) Headspace

The concentration of residual solvent from spray dying in the powder was quantified by GC headspace. A known mass of sample was dissolved in 4 mL of dimethylacetamide in a 20 mL headspace vial. GC was performed on the headspace using an Agilent G7890 (Agilent Technologies, Santa Clara, CA, USA) equipped with a flame ionization detector and split injection capability as well as an Agilent 7697 automated headspace sampler. An Agilent DB-624 column was used with 30 m × 0.32 mm × 1.8 microns. The headspace sampler and instrument parameters used for the experiments are summarized in the Appendix A.

### 2.10. VEGF Binding ELISA Assay

To assess the in vitro binding of spray-dried BEV simul-sprays to VEGF, a Human VEGF Quantikine ELISA kit (Part DVE00, R&D Systems, Minneapolis, MN, USA) was repurposed for a competitive ELISA assay. As a control, the as-received BEV solution was diluted to a 4 mg/mL active in 0.01 M pH 7.4 phosphate buffered saline. Spray-dried powders or as-received API were reconstituted to a 4 mg/mL active in the buffered saline and allowed to dissolve for 60 min. All samples were then centrifuged at 10,000× *g* for 1 min to remove undissolved solids (as would be expected for simul-sprays containing low-solubility actives such as PTX and ERL). Samples were then prepared containing 2.5 nM human recombinant VEGF (R&D Systems, Minneapolis, MN, USA) and 0.75 mg/mL BEV (from spray-dried powder or stock) in the ELISA kit’s calibrator diluent RD5K. For controls which did not contain bevacizumab, identical volumes were transferred. Samples were incubated for 60 min at 37 °C to allow BEV to bind VEGF to equilibrium. Standards were prepared according to the kit protocol.

After incubation, 50 µL of the assay diluent was added to each of the wells of the supplied plate. A total of 200 µL each of the sample (in triplicate) and standards were added to respective wells and incubated for 2 h at ambient temperature. The plate was washed three times with 400 µL of the kit’s wash buffer, then 200 µL of the VEGF-conjugate was added. The plate was incubated for another 2 h, then washed again three times, as previously. A 1:1 mixture of the kit’s color reagent A and B were mixed, then 200 µL was added to each well. After a 30 min incubation at ambient temperature, 50 µL of stop solution was added to each well. The absorbances were read on an M5e plate reader (Molecular Devices, San Jose, CA, USA) using a 450 nm detection wavelength and 540 nm as the blank wavelength for baseline correction. The quantified concentration of unbound VEGF in pg/mL was reported. All samples were analyzed in triplicate.

### 2.11. Solubility

To measure the equilibrium solubility of the API and excipients, a saturated solution of each substance was prepared by stirring the excess solids in the solvent for ~2 h. The solutions were centrifuged to remove excess solids. A total of 50 µL of the supernatant was pipetted into an aluminum pan and placed into a Thermogravimetric Analyzer (Discovery TGA, TA Instruments, New Castle, DE, USA). The sample was heated to 130 °C at a ramp rate of 50 °C/min and then held isothermal for 10 min to allow all of the solvent to evaporate. The final mass of the sample (of the known 50 µL volume) was recorded and used to determine the solubility in mg/mL. Samples were measured in triplicate.

## 3. Results

### 3.1. API and Excipient Solubility

Solubility screening was performed on the three small molecule APIs and the L-leucine excipient in water, 90/10 methanol/water and 80/20 ethanol/water by weight. These solvent ratios were chosen to maximize the solubility of L-leucine and API. Additional details on solubility and formulation selection can be found in the Appendix A. As is evident in Table 3, bevacizumab, erlotinib, and paclitaxel are not soluble in a common solvent at sufficient concentrations. This highlights the need for the simul-spray process to create a combination product. An aqueous buffer can be used for both bevacizumab and cisplatin, but it was found that bevacizumab and cisplatin were not chemically stable when co-dissolved in the same solution.

### 3.2. Residual Solvents

Water content is an important attribute of an inhalable formulation. If water content is too high, the recrystallization of amorphous domains, particularly trehalose, can cause failure. If the water content is near-zero, static can dominate the powder, resulting in poor release from the inhaler device. The water content of the simul-spray SDDs was quantified by Karl Fisher. Overall, all samples contained between 1 and 3.5% water by weight after manufacture.

While the water is retained and important to the formulation, residual organic solvent is undesirable from a safety perspective, and long-term exposure of the BEV to high levels of residual organic solvent could lead to degradation. With these spray conditions, the methanol (ERL-containing powders) and ethanol (PTX-containing powders) were both removed below the ICH limits of 0.3 and 0.5 wt %, respectively.

### 3.3. Drug Concentration

The API concentration in the simul-sprayed powders were quantified on a dry basis. The percent of theoretical concentration is shown for each active in Figure 2. All of the small molecule drug concentrations are within 10% of target. The BEV concentrations are all slightly low, with the ERL 1:1 being 17% below target. A contributing factor to the low concentration of BEV is that proteins can adsorb to the surfaces of the liquid tubing (used to feed the spray dryer), which can lead to aggregation [18]. In our previous work with a mono BEV formulation spray-dried with a similar configuration and formulation, 92.5% of target drug concentration was achieved [17].

A likely cause of the drug concentration discrepancies is the water content differences between the formulations. When measuring drug concentration, the mass of water is subtracted from the total mass of the sample to provide a water-corrected concentration value. This is accomplished by using the bulk residual water content of the powder measured by KF. However, the water is retained differently by the individual formulations. For example, when equilibrated to the same ambient conditions, the weight percent of water for the BEV, ERL, PTX, and CP mono SDDs was 4.7, 0.6, 1.4, and 4.9, respectively. Equilibrated to the same conditions, the simul-sprayed powders show water contents close to what would be expected based on the formulation ratios, but still with some discrepancies, as shown in the Appendix A. Therefore, the BEV values are likely to be slightly under-corrected, especially for the ERL and PTX-containing powders, which could contribute to the low concentration observed in most formulations. Similarly, the small molecule potencies might be slightly over-corrected.

### 3.4. Physical State of Simul-Sprayed Formulations

#### 3.4.1. PXRD

Overall, DSC and PXRD confirmed that the physical state of the simul-spray formulations matches that of the mono formulations. X-ray diffraction was performed on the samples to determine the qualitative presence of crystalline material. In all formulations, characteristic peaks of crystalline L-leucine were found. The presence of crystalline L-leucine is an important formulation attribute for achieving good aerosol properties in spray-dried inhalation dry powders [19]. PXRD confirmed that BEV, PTX, and CP are all amorphous within the instrument’s limit of detection. It also confirmed that ERL was present in crystalline form. A detailed discussion can be found in the Appendix A.

#### 3.4.2. DSC

Thermal analysis was conducted on the simul-spray formulations to demonstrate that the physical state of the materials was not altered by spraying in the presence of a second solvent. All simul-spray formulations showed glass transition temperatures characteristic of the BEV/trehalose phase, with an onset of ~118 °C. Additional glass transitions of the CP/trehalose phase (onset ~109 °C) or PTX phases (onset ~150 °C) were observed in corresponding formulations. Additional details of the thermal analysis can be found in the Appendix A.

#### 3.4.3. Morphology of Particles by SEM

BEV spray-dried formulations and small molecule spray-dried formulations have different surface features and morphology, making it possible to visually distinguish between the particles in SEM images. Exemplary SEM images of this is shown in Figure 3, which depicts BEV, CP, ERL, and PTX mono-API formulations along with their corresponding 1:1 simul-spray formulations. The BEV particles have a smooth surface and are fairly collapsed, while the ERL, PTX, and CP particles have a corrugated surface and are generally closer to spherical in shape. The difference is most prominent for the ERL simul-sprays due to the presence of ERL as crystals.

The differences in surface appearance may be explained by competition for the receding interface of the drying droplet. It has been well-established in the literature that L-leucine-containing inhalation powders should be designed such that the L-leucine crystallizes out of solution during droplet drying, thus enriching on the surface (18). This maximizes its effectiveness as a dispersibility enhancer. It is likely that bevacizumab competes for surface enrichment due to its large molecular size. L-leucine crystal growth could also be delayed due to bevacizumab’s high viscosity near the droplet surface. These factors would lead to a smoother particle surface for the BEV particles as compared with the other APIs.

By visual inspection of the SEM images, it was clear that both components of the simul-sprays have approximately the same particle size distribution. This was then corroborated quantitatively by measurements of the aerodynamic particle size of the powders. Additionally, the images demonstrate the successful and independent atomization of the formulations. No fused particles were observed, confirming minimal to no interactions of the spray plumes before droplet solidification.

### 3.5. Aerosol Properties of Simul-Sprayed Formulations

The aerodynamic particle size distribution of the SDD powders was analyzed using an Aerodynamic Particle Sizer (APS) instrument with a powder disperser. For all formulations, the mass median aerodynamic diameter (MMAD) was between 1.5 and 3 microns, well within the target range of 1–5 microns for lung delivery. The results for MMAD and geometric standard deviation (GSD) are shown in Table 4.

Full particle size distributions for three exemplary formulations are also shown in Appendix A. Notably, the particle size distributions are monomodal in nature, which means that both types of particles within the simul-spray formulations have aerodynamic diameters that are in-range for pulmonary delivery.

A fast-scanning impactor (FSI) was used to quantify the fine particle dose (FPD) normalized by the capsule fill mass (FPD/fill mass). The FSI is similar to the more popular Next Generation Impactor or the Anderson Cascade Impactor, but a single stage is used to gravimetrically quantify powder with an aerodynamic diameter of <5 microns. A capsule containing 10 mg of the SDD powder was loaded into a dry powder inhaler for use with the impactor. A summary of the data for this test is shown in Table 4. Overall, PTX simul-sprays had the highest %FPD/nominal, and ERL had the lowest.

To see if the formulations uniformly aerosolized, the FPD portion from the FSI testing of select samples were analyzed for drug concentration of the actives. Similar FPD/Fill mass values were found for both actives in each case, and more details are provided in the Appendix A.

### 3.6. Anti-VEGF Activity of BEV Simul-Sprays

An ELISA-based binding assay was used to assess the ability of BEV in the simul-sprayed powders to disrupt VEGF binding. Briefly, reconstituted simul-spray powders were incubated with VEGF protein in the assay buffer. Next, the quantity of unbound VEGF in the solution was quantified by a commercially available VEGF ELISA assay kit. The lower the quantified level of VEGF from the assay, the more active BEV was at binding VEGF. As-received BEV stock solution was used as a positive control for the assay. Four negative controls were demonstrated: a solution containing VEGF and no actives, as well as VEGF/ERL, VEGF/PTX, and VEGF/CP solutions without BEV. These were used to demonstrate that the small molecules did not interfere with VEGF binding on their own.

Figure 4 shows the results of the VEGF assay. The positive control sample, using BEV stock, has 318 pg/mL of unbound VEGF. The negative control samples all read VEGF values of >3000 pg/mL, saturating the detection of the assay, as expected. The ERL, PTX, and CP negative controls all show that the small molecules do not inhibit VEGF measurably. Both the ERL 1:2 and 1:1 simul-sprays have similar values, which are the same as those for the positive control within the error of the assay. All PTX formulations show VEGF-inhibition levels similar to the positive control. All three cisplatin simul-sprays show VEGF inhibition. It is not known at this time why the CP 2:1 formulation appears to inhibit more strongly than the others, and detailed assay development is out of the scope of this work.

Overall, the activity assay confirmed that active BEV is present in the simul-sprays, with the preserved ability to inhibit VEGF after spray drying indicating its preserved biologic activity. This finding is similar to that of our previous work with spray-dried BEV monotherapies, where VEGF activity was quantified using a cell-based assay [17].

## 4. Discussion

### 4.1. Advantages of Simul-Spray

The simul-spray technique demonstrated in this work provides a platform for the manufacture of diverse combination powders with matched aerosol properties. In particular, it facilitates combination therapies with one or more of the following challenges:API sensitivity to shear from milling processes or otherwise is not suited to milling;The need for a high dose is incompatible with a carrier-based DPI technology;There are APIs which cannot be dissolved in a common volatile solvent for spray drying.

Here, we give an overview of how simul-spray processing can be leveraged to overcome these challenges.

#### 4.1.1. Ease of Formulation Optimization

As demonstrated above, APIs without common solvents are a key area where simul-spray can facilitate the combination of products. With the need for a common solvent eliminated, formulations can be independently optimized to suit each active and then simul-sprayed. This provides great flexibility in excipient selection as well as the option to “re-formulate” by simply changing the ratio of the feed solutions. For a combination product, this could be particularly valuable during clinical trials, when the complicated dose optimization process is still ongoing. For highly potent compounds, simul-spray could also be used to perform dose escalation studies in which the second formulation is a placebo. This would eliminate the need for different fill weights or multiple actuations in a clinical trial. In many ways, the use of the simul-spray process could decrease the amount of formulation work needed to develop a combination inhalation dry powder.

For the case studies demonstrated in this work, two of the small molecule APIs are insoluble in water (ERL and PTX) and thus cannot be mixed with the BEV solution. BEV is unstable in organic solutions. Cisplatin has slight solubility in water, but it is not chemically stable in an aqueous solution for longer than a few hours as it is susceptible to ligand exchange [20,21]. Thus, simul-spray was particularly enabling for the model compounds demonstrated here.

#### 4.1.2. Overcoming Manufacturing Concerns

A primary concern with simul-spray manufacturing is that during manufacturing, differences in powder buildup on the dryer walls between the two formulations could lead to composition change. Likewise, differences in particle size could cause the cyclone to preferentially bypass the formulation with smaller particles. Thus, verification of the target drug concentration was critical to demonstrating the simul-spray technique. While not exactly matched with desired target values, the results shown in this proof-of-concept work establish the feasibility of the technique by these metrics. The optimization of the spray drying parameters as well as individual formulations and particle size distributions would lead to more robust product profiles, but these were out of the scope of this study.

SEM and aerosol characterization techniques confirmed that respirable powder properties were achieved for both formulations within each simul-spray. The aerosol properties of the powders, measured by APS and FSI, were consistent with targeted delivery to the deep lung. This further emphasized that the non-biased collection of the two formulations occurred during manufacturing. An evaluation of the active concentrations of the FSI fine particle dose showed that the formulation ratios are maintained during aerosolization. Additionally, the BEV retained its anti-VEGF bioactivity, as demonstrated by the ELISA quantification. This was of particular interest for ERL and PTX simul-spray formulations, in which the BEV might have been impacted by ethanol or methanol vapor exposure. Altogether, these results allay the major concerns about simul-spray manufacturing.

Though not explored in this study, combination formulations with deliberately different aerosol properties could be prepared to expand the therapeutic range of the actives. For example, one formulation could be manufactured with smaller particle size to target the alveolar region of the lung, while a second could be sized to target the conducting airways. This approach would not be without its engineering challenges, particularly around the cyclonic collection of powders with diverse aerodynamic properties.

#### 4.1.3. Avoid Blending and Carrier Particles

Blending poorly flowing inhalation dry powders is a substantial challenge in many cases. Once simul-spray drying is complete, no further blending operations are necessary, and the powder can be filled directly into blisters or capsules without the need for additional excipients.

For a biotherapeutic such as the mAb used in this study, milling is not an option. Materials are typically supplied as liquid solutions, and even when lyophilized, shear sensitivity is a challenge, which would preclude milling. Even for compounds where milling is feasible, spray drying may still be a preferred particle engineering technology for inhalation delivery as it allows for the manufacture of formulations without the use of inert carrier particles. This is particularly helpful when high doses (e.g., >5 mg) are required for treatment. For high-dose compounds, eliminating the need for carrier particles can help reduce the need for multiple actuations of the dry powder inhaler, which is a high burden for cystic fibrosis patients, for example [14].

### 4.2. Significance of the Model Systems

The model systems chosen for this study were selected due to their potential relevance to the treatment of lung cancer. BEV is a VEGF-inhibitor monoclonal antibody first marketed as Avastin [16]. BEV is approved for the treatment of late-stage non-small-cell lung cancer (NSCLC). It is administered intravenously, often in combination with chemotherapy, immunotherapy, or other targeted therapies such as EGFR-inhibitors [22,23]. Our recent study on an inhaled formulation of BEV manufactured by spray drying demonstrated the successful reduction of tumors in a rat model for NSCLC [17].

Compounds to pair with BEV for this simul-spray drying proof-of-concept study were inspired by a review of inhaled chemotherapy by Rosiere et al. [24]. This work highlighted the potential of dry powder inhalers to deliver chemotherapeutic agents directly to the lung, circumventing many of the safety challenges of nebulizer delivery. To this end, two chemotherapeutic APIs were chosen as model compounds: PTX and CP. In addition, the EGFR-inhibitor ERL was selected as a third model system, which could potentially be relevant to patients whose tumors have EGFR mutations.

From a therapeutic standpoint, the local treatment of lung disease, particularly of lung cancer, using dry powder inhalers specifically has many potential patient benefits, including:Reduced dose;Reduced systemic side effects;Avoidance of cold chain storage requirements;Simple, at-home administration;Reduced cost of treatment.

These advantages are discussed in greater detail in recent publications [17,25,26] and review articles [27]. The combination therapies exemplified were chosen from approved cancer therapies and were intended to serve as a proof of concept which could help patients who are dealing with a challenging disease. For chemotherapy in particular, local treatment by inhalation has many remaining hurdles to its implementation, although dry powder inhalers have the potential to address some of these issues [24]. In the current standard of care, a late-stage lung cancer patient receives both CP and BEV by IV infusion at recurring in-clinic appointments. Although this work is at an early stage, the vision of having a single treatment administered non-invasively at home by dry powder inhaler is a compelling one for patients.

## 5. Conclusions

This article demonstrated the simul-spray drying technology in which spray-dried particles of two different compositions are atomized into the dryer simultaneously with the use of two separate nozzles, forming a uniform powder blend. Simul-spraying was used in this work to manufacture combination therapies for inhalation which contain BEV and small-molecule cancer therapies: ERL, PTX, and CP. The resulting powders achieved their target drug concentration, good aerosol properties for delivery to the lung, and preserved anti-VEGF bioactivity. For lung cancer, inhaled combination therapies could locally treat this complex disease, easing patient compliance, reducing the dose, and limiting the exposure of healthy tissue to toxic compounds. More generally, a simul-spraying technique can be used to prepare combination inhalation therapies with otherwise incompatible APIs. The process could eliminate additional blending operations on poor-flowing inhalation powders and potentially shorten development timelines for combination inhalation products.

## Figures and Tables

**Figure 1 pharmaceutics-14-01130-f001:**
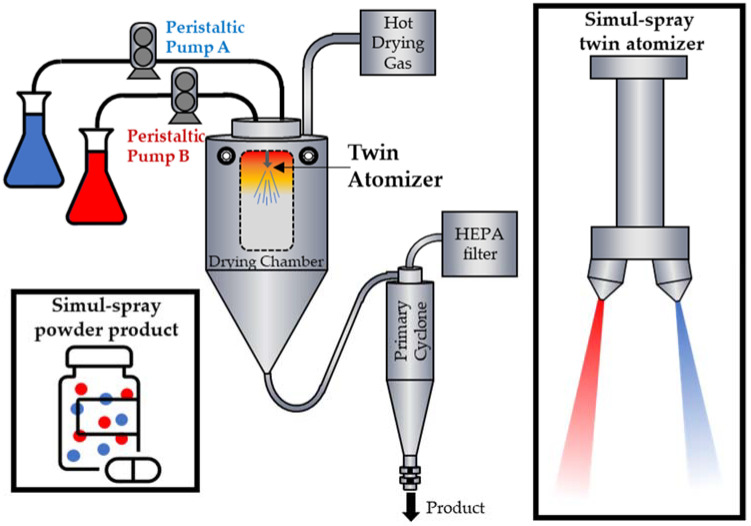
Simul-spray drying setup schematic. For simplicity, the atomization gas supply lines are not depicted.

**Figure 2 pharmaceutics-14-01130-f002:**
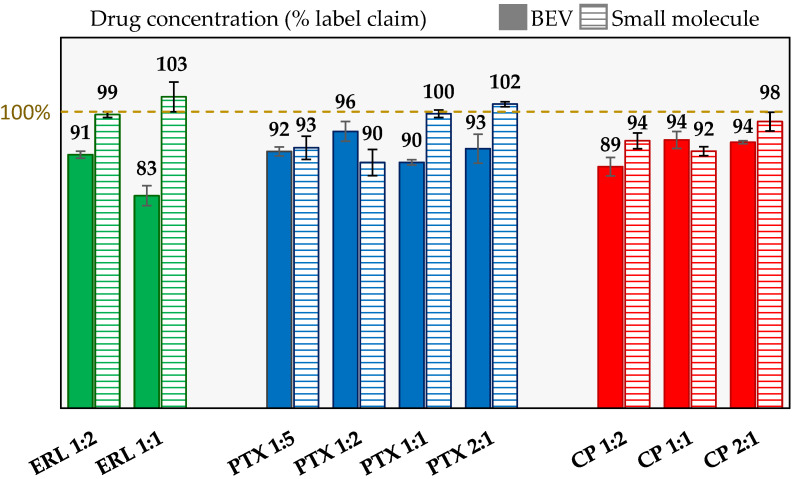
Drug concentration, listed as a percent of theoretical, of the active components for each of the simul-spray-dried powders, with solid-filled bars showing the value for the small molecules and the line-filled bars showing the BEV concentrations.

**Figure 3 pharmaceutics-14-01130-f003:**
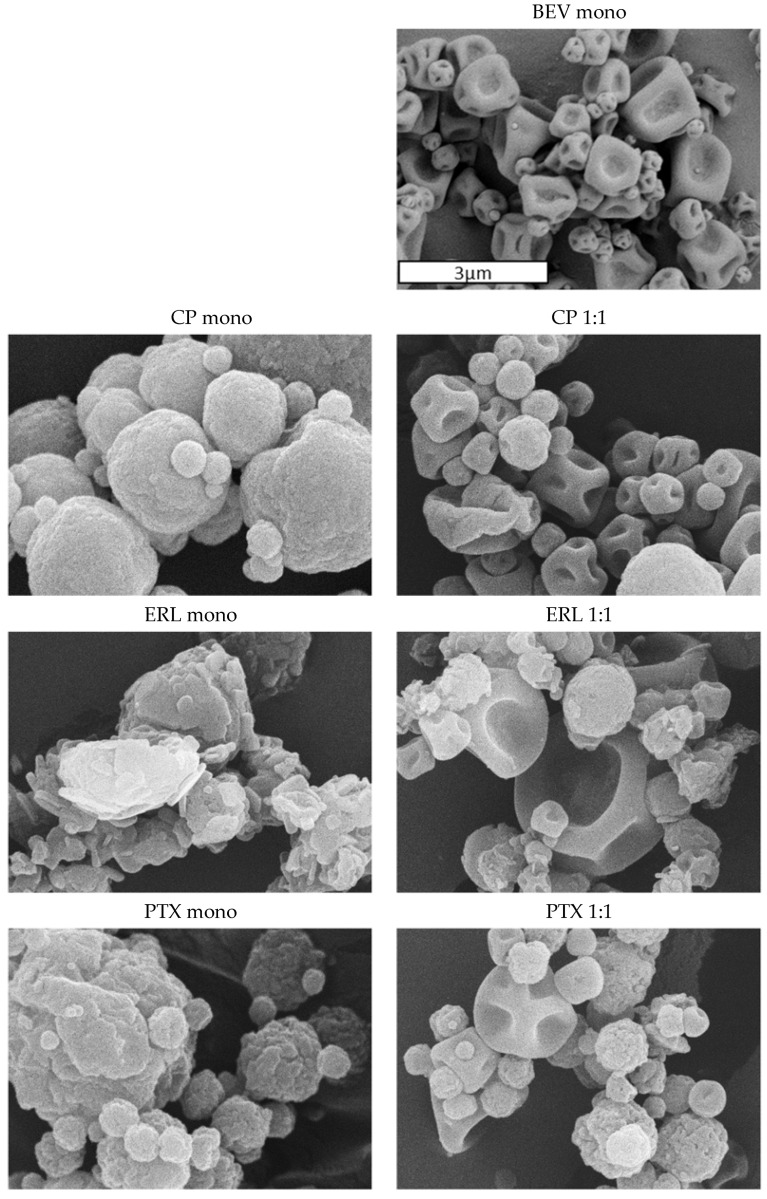
Representative SEM images of mono-API and 1:1 simul-spray formulations. The scale bar is the same for all images.

**Figure 4 pharmaceutics-14-01130-f004:**
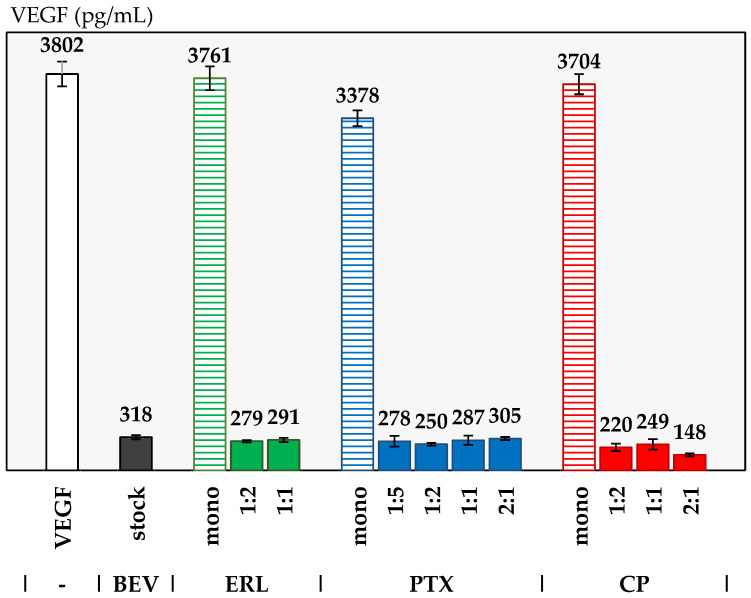
VEGF activity in pg/mL of the mono API controls (dashed) compared to the VEGF blank (white), and of the simul-spray formulations (solid) compared to the BEV control (black).

**Table 1 pharmaceutics-14-01130-t001:** Summary of individual formulation spray solutions (compositions are by mass).

Individual SDD ID	Formulation	Spray Solvent	Solution Concentration (mg/mL)
(A)	40/40/20 BEV/trehalose/L-leucine	1 mM phosphate buffer, pH 6.3	10
(B)	80/20 ERL/L-leucine	90/10 methanol/water	10
(C)	80/20 PTX/L-leucine	80/20 ethanol/water	7.5
(D)	10/70/20 CP/trehalose/L-leucine	DI water	10

**Table 2 pharmaceutics-14-01130-t002:** Summary of formulations used in the study, with active loading compositions and manufacturing liquid flow rates.

Mixed SDD Formulation ID	Individual SDD Ratios by Mass ^1^	Solution Flow Rates (g/min)		Powder API Content (wt %)	
		Small Molecule	BEV	Small Molecule	BEV
BEV mono	(A) only	NA	6.0	0	40
ERL mono	(B) only	6.0	NA	80	0
ERL 1:2	(B):(A) 1:2	2.0	4.0	26.7	26.7
ERL 1:1	(B):(A) 1:1	3.0	3.0	40	20
PTX mono	(C) only	6.0	NA	80	0
PTX 1:5	(C):(A) 1:5	1.5	5.0	13.3	33.3
PTX 1:2	(C):(A) 1:2	3.0	4.0	26.7	26.7
PTX 1:1	(C):(A) 1:1	3.4	2.6	40	20
PTX 2:1	(C):(A) 2:1	6.1	2.0	53.3	13.3
CP mono	(D) only	6.0	NA	10	0
CP 2:1	(D):(A) 2:1	4.0	2.0	6.7	13.3
CP 1:1	(D):(A) 1:1	3.0	3.0	5	20
CP 1:2	(D):(A) 1:2	2.0	4.0	3.3	26.7

^1^ Formulation information for (A–D) found in Table 1.

**Table 3 pharmaceutics-14-01130-t003:** Summary of solubility in selected spray solvents (ratios are by mass). NT = not tested.

Component	mg/mL in Water	mg/mL in 90/10 Methanol/Water	mg/mL in 80/20 Ethanol/Water
BEV	>100	NT; incompatible	NT; incompatible
ERL	<1	25	<1
PTX	<1	<1	8.0
CP	2.5	<1	<1
L-Leucine	21	2.5	2.0

**Table 4 pharmaceutics-14-01130-t004:** Aerosol properties of simul-sprayed formulations: MMAD and GSD by APS, and %FPD/nominal by FSI.

Formulation	APS MMAD (µm)	APS GSD (µm)	FSI FPD/ Fill Mass, %
ERL 1:2	2.9 ± 0.3	1.7 ± 0.1	43.4 ± 2.5
ERL 1:1	2.5 ± 0.7	1.7 ± 0.1	46.3 ± 1.5
PTX 1:5	2.3 ± 0.02	1.6 ± 0.01	64.3 ± 8.0
PTX 1:2	2.4 ± 0.4	1.7 ± 0.1	64.0 ± 0.0
PTX 1:1	2.4 ± 0.3	1.7 ± 0.1	54.6 ± 7.0
PTX 2:1	1.8 ± 0.1	1.7 ± 0.03	65.2 ± 5.9
CP 1:2	2.8 ± 0.02	1.7 ± 0.03	58.0 ± 0.7
CP 1:1	2.7 ± 0.01	1.7 ± 0.01	57.7 ± 1.6
CP 2:1	2.7 ± 0.03	1.7 ± 0.01	59.9 ± 2.7

## Data Availability

Not applicable.

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
