# Peer review of "Simultaneous Spray Drying for Combination Dry Powder Inhaler Formulations"

_pharmaceutics, 2022, doi:10.3390/pharmaceutics14061130_

Round 1

Reviewer 1 Report

In this work, a simultaneous spray drying process, termed simul-spray, is described where two different active pharmaceutical ingredient solutions are simultaneously atomized through separate nozzles into a single spray dryer. Valuable work with novelty.

However, the presentation of the results is difficult to follow. The labeling of active substances is inconsistent e.g. PTX in one place and Paclitaxel in the other, it is confusing to change the order of the tested samples e.g. a FIG. 3 or in the annex once CP, ERL, PTX, other times PTX, ERL, CP.

The method part does not contain a description of the determination of the degree of crystallinity, however, the supplementary contains data on this. In connection with this, the LOD value must also be replaced. The GSD parameter should also be defined (see Table 4) in method part.

The value of the work would be increased if the results of the ICH stability testing of the developed products were included in the manuscript, especially with regard to the changes in the amorphous and crystalline structure of the components.

Author Response

We thank the reviewer very much for their thorough and detailed review of our manuscript. 

  1. We modified the manuscript throughout to be more clear with the labelling of active substances.
  2. The method section was modified to include information on quantifying the percent crystallinity, as well as the GSD.
  3. Your comments on the value of a stability study are well-taken. We agree that this would increase the value of the manuscript, though unfortunately we were not able to conduct this work at this time. For the bevacizumab-only spray dried powders, we have previously published stability data showing at least 6 month stability at 25C (see Shepard et al reference in this manuscript).

Reviewer 2 Report

The idea of simultaneous spray drying of APIs together in single spray dryer is insightful; however, this manuscript needs some grammatical and experimental changes before its consideration for publication.

Comments:

  1. In abstract, reviewer suggests to check the grammatical mistakes and also try to represent it in more meaningful manner. For instance, author mentioned that “which requires a high dose incompatible with carrier-based formulation”, this line is either incomplete or can be presented in another way.
  2. In introduction portion, abbreviation of COPD is missing. Please mention its full form initially.
  3. In materials and methods and results also, authors mentioned word potency. how potency can be measured by HPLC? Authors used HPLC for determination of active ingredients in resultant spray dried powder inhaler, which can be mentioned as drug loading/drug content/encapsulation rather potency.

Author Response

Thank you very much for your detailed review of our manuscript. 

We will correct the grammatical errors you pointed out, and also switch the word "potency" to "drug concentration," as your suggestion is correct.

Reviewer 3 Report

The manuscript entitled “Simultaneous spray drying for combination dry powder inhaler formulations” reports the spray drying of two active pharmaceutical ingredients, with different physicochemical characteristics.

Potency was evaluated by HPLC, drug stability by DSC, particle size of the powders, morphological characterization of the particles, solubility assays, ELISA Assay, solvent residue analysis.

Thus, the researchers demonstrated that the technique allows obtaining powders with a particle size suitable for administration, with adequate concentrations of the two APIs, without affecting the stability of the drug and with a potency that allows compliance with the dosage requirements.

In relation to the writing of the manuscript, the title is clear, the abstract is adequate, since it gives a clear idea of ​​the content of the manuscript. Regarding the introduction, it allows identifying the need for the evaluated technique.

The methodology is clear and precise, which would allow replication of the study by other researchers. The discussion of the results is scientifically sound, the authors manage to connect each of their results and support them with a bibliography that consolidates the discussion of the data obtained.

The analytical techniques used are well described and allow obtaining high quality experimental data.

This was a very interesting reading and the quality of the manuscript is substantial. I recommend its publication as is.

Author Response

Thank you very much for the review of our work and for your recommendation to publish as-is. 

Round 2

Reviewer 1 Report

The corrected manuscript is suitable for publication.